# *Toll-like Receptor 9* Gene in the Development of Type 2 Diabetes Mellitus in the Saudi Arabian Population

**DOI:** 10.3390/biology12111439

**Published:** 2023-11-16

**Authors:** Zeina S. Alkudmani, Aminah Ahmad Alzailai, Khaled H. Aburisheh, Amal F. Alshammary, Imran Ali Khan

**Affiliations:** 1Department of Clinical Laboratory Sciences, College of Applied Medical Sciences, King Saud University, Riyadh 11433, Saudi Arabia; zalkudmani@ksu.edu.sa (Z.S.A.); amalzailai@ksu.edu.sa (A.A.A.); aalshammary@ksu.edu.sa (A.F.A.); 2University Diabetes Center, King Saud University Medical City, King Saud University, Riyadh 11472, Saudi Arabia; kaburisheh@ksu.edu.sa

**Keywords:** diabetes mellitus, type 2 diabetes mellitus/T2DM, *TLR9* gene, rs187084, rs352140, rs5743836, Saudi Arabia

## Abstract

**Simple Summary:**

The *TLR9* gene is known to be a major element in an innate immune system. The relation between TLR9 and diabetes is connected via chronic inflammation, which affects the β-cell function. Previous studies on diabetes were with rs187084, rs352140, and rs5743836 SNPs in the *TLR9* gene, and there are no direct studies. The prevalence of T2DM was expanding in Saudi Arabia. One of the foot ulcer infections in diabetic individuals is linked to the relationship between infection and diabetes. Individuals who had a COVID-19 infection are at an increased risk of developing chronic diseases, such as diabetes; this has been observed in Saudi Arabia. We design this study between rs187084, rs352140, and rs5743836 SNPs in the *TLR9* gene and T2DM in the Saudi population. The current study results confirmed rs187084 and rs5743836 SNPs were associated in the Saudi population. This indicates that they play a role in the Saudi population.

**Abstract:**

Diabetes mellitus is a complex disease with a wide range of manifestations. Diabetes, notably type 2 diabetes mellitus (T2DM), is becoming more common in Saudi Arabia as a result of obesity and an aging population. T2DM is classified as a noncommunicable disease, and its incidence in the Saudi population continues to grow as a consequence of socioeconomic changes. Toll-like receptors (TLRs) are innate immune receptors that mediate the inflammatory response in diabetes mellitus. Previous studies have documented the relationship between different SNPs in the *TLR9* gene in different forms of diabetes. As a result, the purpose of this study was to investigate the relationship between rs187084, rs352140, and rs5743836 SNPs in the *TLR9* gene among T2DM patients in the Saudi population. This was a case-control study that included 100 T2DM cases and 100 control subjects. The three SNPs were identified in the study population (n = 200) using polymerase chain reaction (PCR), restriction enzymes for rs352140, and Sanger sequencing for rs187084 and rs5783836. Next, statistical analyses were performed using various software to determine the association between the SNPs and T2DM. rs187084 and rs5743836 were associated with an increased risk of T2DM development. rs187084 and rs5743836 allelic frequencies were associated with a 3.2 times increased risk of T2DM development (*p* < 0.05). DBP was associated with T2DM (*p* = 0.02). rs187084 was associated with TC and HDLc; rs352140 was associated with DBP, HbA1c, and HDLc; rs5743836 was associated with waist (*p* < 0.05). The CGT haplotype was strongly associated with T2DM (*p* < 0.003). Gene–gene interaction, graphical presentation, and dendrogram showed the strong association with T2DM patients (*p* < 0.05). This study concluded that rs187084 and rs5743836 were strongly associated with T2DM in Saudi Arabian patients. This study provides further evidence that SNPs in the *TLR9* gene play a significant role in T2DM development in a Saudi community.

## 1. Introduction

Globally, diabetes mellitus (DM) is one of the most common chronic diseases, and its prevalence has increased significantly over the last three decades [1]. At present, 460 million individuals have DM; this number is expected to increase to 700 million by 2045. It is a chronic and metabolic condition characterized by elevated blood glucose levels [2]. The three most common types of diabetes are type 1 diabetes mellitus (T1DM), type 2 diabetes mellitus (T2DM), and gestational diabetes mellitus (GDM). According to WHO, Saudi Arabia has the seventh highest diabetes rate in the world and the second highest rate in the Middle East, with an estimated 7 million diabetes and 3 million individuals with diabetes [3]. Saudi Arabia is one of the 21 countries and territories of the International Defense Force (IDF) Middle East and North Africa (MENA) Region. Of the 537 million individuals with diabetes worldwide, 73 million are in the MENA Region; this number is expected to increase to 135.7 million by 2045 [4]. Currently, diabetes is the most prevalent serious health concern in Saudi Arabia. According to the Ministry of Health in Saudi Arabia, the prevalence of diabetes increased from 0.9 million to 2.5 million between 1992 and 2010. However, the prevalence is expected to increase 2.7 times within the next two decades [5].

Type 2 diabetes mellitus (T2DM) is characterized by hyperglycemia and is caused by a combination of deficient insulin secretion, inadequate insulin resistance, and misleading glucagon secretion [6]. Since 1960, the prevalence of T2DM has increased in tandem with obesity. T2DM is caused by a combination of genetic and lifestyle factors [7]. Ethnicity, age, body mass index (BMI), obesity, blood pressure, fasting blood glucose (FBG), and lipid concentrations have all been identified as risk factors for T2DM development [8]. T2DM has been linked to several gene variants and a number of single nucleotide polymorphisms (SNPs; monogenic and polygenic) forms [9]. SNPs were discovered through the human genome sequence in 2003, and genome-wide association studies (GWAS) have detected associations between disease and the genetic loci [10].

Toll-like receptors (TLRs) play an important role in the activation of innate immunity and regulatory adaptive immunological responses. TLRs, in humans, are classified into ten subtypes [11]. Effector molecules, such as cytokines and chemokines, regulate adaptive immune responses and are expressed when TLRs recognize pathogen-associated molecular patterns [12]. TLR9 is linked to autoimmune diseases and has been associated with high-risk asthma and low-risk diabetes [13]. TLR9 deficiency enhances pancreatic islet formation as well as β-cell differentiation and function [14]. Bizarrely, TLR9 deficiency was found to boost glucose tolerance and insulin sensitivity in individuals with diabetes. Han et al. have confirmed a connection between TLR9 and T2DM osteoporosis via the NF-_K_B pathway [15]. One of the foot ulcer infections in individuals with diabetes is linked to the relationship between infection and diabetes [16].

The relationship between *TLR9* and diabetes, specifically T2DM, is well established [12,13,17]. The rs352140 SNP was studied in T1DM, and it was confirmed that the rs352140 SNP was connected with residual β-cell function [11]. There are no direct studies between rs187084 SNP and T2DM, but Liu et al. [17] carried out studies on the CAD individuals who developed T2DM. Finally, the rs5743836 SNP was studied in individuals with T2DM and diabetic foot [12]. The main purpose of designing this study specifically in T2DM patients was due to the documentation of the previous studies with these three SNPs in different human diseases in Saudi Arabia [18,19]. Altogether, the relation between the *TLR9* gene and T2DM was connected to β-cell function, and additionally, CD73 expression was significantly upregulated and the immunosuppressive function of CD73+ T cells was improved in NOD mice with a TLR9 deficit. As a result, the development of diabetes was slowed while the levels of proinflammatory cytokines were reduced and the expression levels of more anti-inflammatory cytokines were elevated [20]. In this study, we aimed to determine the relationship between three TLR9 SNPs, namely rs187084, rs352140, and rs5743836, and T2DM. We aimed to determine the molecular role of rs187084, rs352140, and rs5743836 in Saudi Arabian patients, with a familial history of diabetes, diagnosed with T2DM.

## 2. Materials and Methods

### 2.1. Ethics

Ethical approval and funding were sanctioned by the King Saud University (KSU) Institutional Review Board. Written informed consent was obtained from all patients who participated in this study. This study was carried out in compliance with the Saudi Ministry of Health’s principles of ethics and the Helsinki Declaration.

### 2.2. Participants and Research Settings

We conducted a case-control study in the G-141 laboratory at the KSU Department of Clinical Laboratory Sciences, College of Applied Medical Sciences. In this study, 100 T2DM and 100 healthy control samples were collected from Saudi Arabian patients who visited KSU-associated diabetes clinics based on common criteria. All T2DM patients (n = 100) had a family history of diabetes on either the paternal or maternal side, whereas healthy controls (n = 100) did not have a family history of diabetes. The T2DM inclusion criterion was based on the WHO cut-off value for the diagnosis of diabetes, that is, fasting blood glucose (FBG) > 7.0 mmol/L. Participants with FBG levels < 7.0 mmol/L were excluded. Healthy controls were confirmed based on the same FBG cut-off value, and healthy control participants with FBG > 7.0 mmol/L were excluded from the control group. Patients who are with mixed population were also excluded from this study in T2DM cases and controls. The sample size formula was adopted from a couple of documented studies [21,22]. The following formula was applied in our study for a case-control study.
$$n=\left(Z_{\alpha /2}+Z_{\beta }\right)2^{2}{}^{*}2^{*}{}^{62/d2}$$

### 2.3. Parameters of Study Participants

In this study, anthropometric and clinical details were collected from all participants. Age, weight, height, BMI, waist, hip, systolic blood pressure (SBP), diastolic blood pressure (DBP), and family history of diabetes were recorded.

### 2.4. Blood Collection

Following overnight fasting (minimum of 10–12 h), 5 mL of peripheral blood was collected. A total of 2 mL of the blood was collected in a plain vacutainer for biochemical analysis, 2 mL in an EDTA vacutainer for molecular analysis, and 1 mL in an EDTA vacutainer for HbA1c analysis.

### 2.5. Serum Analysis

The 2 mL blood sample collected in the coagulant tube was centrifuged to obtain the blood serum. Fasting blood glucose (FBG) and lipid profile parameters (total cholesterol (TC), triglycerides (TG), high density lipoprotein cholesterol (HDLc), and low density lipoprotein cholesterol (LDLc)) were measured in the serum.

### 2.6. DNA and RFLP–Sanger Sequencing Analysis

Genomic DNA was extracted from the blood contained in EDTA (200 µL) using the Qiagen kit, according to the manufacturer’s instructions. The extracted DNA was quantified using a NanoDrop spectrophotometer, and the DNA samples were adjusted to 20 ng/µL. PCR was used to genotype rs187084 (−1486T>C), rs352140 (G2848A), and rs5743836 (−1237T>C) in the DNA samples (n = 200). The PCR was performed using the extracted DNA as the template (20 µg/mL) and forward/reverse primers (10 pmol/µL) using the Qiagen PCR Master mix, according to the manufacturer’s instructions. The PCR was performed in a thermocycler (Applied Biosystems, Waltham, MA, USA) using the following conditions: initial denaturation at 95 °C for 5 min, 35 cycles (denaturation at 95 °C for 30 s, annealing at 60 °C/68 °C for 30 s, extension at 72 °C for 45 s), and final extension at 72 °C for 5 min. The rs352140 SNP PCR products were digested with BstUI NEB restriction enzyme for 18 h at 37 °C. PCR and RFLP products were verified by electrophoresis on 2% and 3% agarose gels, using ethidium bromide (10 ng), and an image was captured using a UV-gel documentation system (Figure 1). The NEB cutter did not detect restriction sites within the rs5743836 and rs187084 SNP sequences; therefore, Sanger sequencing was used to determine the presence of these SNPs in the 100 T2DM and 100 healthy control samples (Figure 2). Simultaneously, 9% of the rs352140 SNP samples were validated using Sanger sequencing. Sanger sequencing was performed as previously described. Table 1 contains complete details of all the SNPs used in this study, including primers, PCR–RFLP, and band sizes.

### 2.7. Statistical Analysis

Categorical and numerical data were documented as mean ± standard deviation and total number (percentages). T2DM cases and control baseline characteristics were compared using SPSS (version 27.0, USA; Table 2). In addition, using SNPSTAT, we performed Hardy Weinberg equilibrium (HWE) analysis (Table 3), genotype frequency analysis (Table 4), and allele frequency analysis (Table 5). Multiple logistic regression analysis was performed between T2DM covariates and three SNPs (Table 6) using SPSS software. One-way ANOVA analysis (Table 7) was performed between T2DM covariates and the three SNPs with Kruskal–Wallis tests using Jamovi software (version 2.3.21). Haplotype analysis was performed using ORs, 95% CI, and *p*-values using SNPSTAT software (Table 8). Linkage disequilibrium (LD) analysis (Figure 3) was performed using Haploview software (Version 4.2) to determine the D’ values of the three SNPs present in the *TLR9* gene (Table 9). Multifactor dimensionality reduction (MDR) and generalized multifactor dimensionality reduction (GMDR) analysis were applied with gene–gene interaction (Table 10). Figure 4 represents the light and dark cells and Figure 5 was studied for dendogram analysis. A *p*-value ≤ 0.05 was considered statistically significant.

## 3. Results

### 3.1. Clinical Analysis of Enrolled Participants

The demographic details of all study participants (n = 200) are summarized in Table 2 and include anthropometric, biochemical, and clinical details. In this study, all individuals were natives of Saudi origin. The mean age of the T2DM group (63.27 ± 8.37) was higher than that of the control group (52.12 ± 7.31). There was a higher number of males in the T2DM group than the control group (63% vs. 54%, respectively). Waist size in the control group was larger (97.64 ± 18.68) than that in the T2DM group (96.58 ± 17.12). Weight (79.88 ± 13.79; *p* = 0.05), hip (104.18 ± 16.11; *p* = 0.81), TG (2.11 ± 1.99; *p* = 0.19), and LDLC levels (3.80 ± 0.98; *p* = 0.34) were higher in the T2DM group and showed no association. BMI (31.30 ± 5.18; *p* = 0.004), SBP (126.30 ± 10.75; *p* < 0.0001), DBP (79.76 ± 6.42; *p* = 0.002), FBG (12.99 ± 4.38; *p* < 0.0001), HbA1c (7.17 ± 0.87; *p* < 0.0001), TC (5.76 ± 1.20; *p* = 0.001), and HDLC (1.37 ± 1.20; *p* < 0.0001) were elevated in the T2DM group and showed positive association (*p* < 0.05). The T2DM group had 100% family history of diabetes, whilst the control group had 43% documented family history of diabetes (*p* < 0.0001).

### 3.2. Analysis of HWE Parameter

In this study, we have selected three SNPs in the *TLR9* gene. HWE analysis of the selected SNPs was performed (Table 3). rs187084 and rs5743836 SNPs were approved (*p* < 0.05), and rs352140 was found to be inconsistent with HWE analysis (*p* = 0.25). Natural selection over a specific genotype, resulting in unequal fitness, is a common cause of HWE disruptions.

### 3.3. Analysis of Different SNP Genotypes Present in the TLR9 Gene

The quality of genomic DNA was found to be between 1.7 and 2.0 with an estimated OD260/OD280 ratio. Table 4 represents genotype frequencies of rs187084, rs352140, and rs5743836 SNPs in the T2DM and control groups. The rs187084 SNP, confirmed as TT, TC, CC genotypes, was found in the T2DM group (TT: 49%, TC: 28%, CC: 23%) and the control group (TT: 78%, TC: 14%, and CC: 8%). TC and CC genotypes in the T2DM group were higher than those in the control group, and statistical analysis confirmed positive association with genotypes (TC vs. TT: OR—3.18 [95% CI: 1.52–6.63]; *p* = 0.001 and CC vs. TT: OR—4.57 [95% CI: 1.89–11.04]; *p* = 0.0003) and the dominant model (TC + CC vs. TT: OR—3.69 [95% CI: 1.99–6.82]; *p* = 0.0002). However, other genetic models (TT + CC vs. TC: OR—0.41 [95% CI: 0.20–0.85]; *p* = 0.01 and TT + TC vs. CC: OR—0.29 [95% CI: 0.12–0.68]; *p* = 0.003) showed negative associations.

The rs352140 SNP, presented as GG, GA, and AA genotypes, was found in the T2DM group (GG: 32%, GA: 43%, and AA: 57%) and the control group (GG: 18%, GA: 59%, and AA: 23%). Statistical analysis confirmed a positive association between genotypes (GA vs. GG: OR—0.41 [95% CI: 0.21–1.82]; *p* = 0.01 and AA vs. GG: OR—0.61 [95% CI: 0.27–1.37]; *p* = 0.23); however, there was no association between different genetic models (GA + AA vs. GG: OR—0.46 [95% CI: 0.24–0.91]; *p* = 0.02 and GG + GA vs. AA: OR—0.89 [95% CI: 0.46–1.71]; *p* = 0.74). There was a positive association in the co-dominant model, i.e., GG + AA vs. GG: OR—1.91 [95% CI: 1.08–3.34]; *p* = 0.02. GA and AA genotypes were high in control as well as T2DM patients. This indicates that the rs352140 SNP has a protective effect in T2DM patients.

The rs5743836 SNP, present as TT, TC, and CC genotypes, was found in the T2DM group (TT: 74%, TC: 15%, and CC: 11%) and the control group (TT: 91%, TC: 5%, and CC: 4%). Heterozygous (TC) and homozygous variants (TC) were highly prevalent in the T2DM group. Genotype analysis confirmed positive association between genotypes (TC vs. TT: OR—3.68 [95% CI: 1.28–10.62]; *p* = 0.01 and CC vs. TT: OR—3.38 [95% CI: 1.03–11.07]; *p* = 0.03) and the dominant model (TC + CC vs. TT: OR—3.55 [95% CI: 1.56–8.04]; *p* = 0.001). However, other genetic models (TT + CC vs. TC: OR—0.29 [95% CI: 0.10–0.85]; *p* = 0.01 and TT + TC vs. CC: OR—0.33 [95% CI: 0.10–1.09]; *p* = 0.06) showed negative associations.

### 3.4. Analysis of Allele Frequencies Present in TLR9 Gene SNPs

The differences between the three SNPs present in the *TLR9* gene are shown in Table 5. rs187084 SNP C and T alleles were found in C: 37%/T: 63% and C: 15%/T: 85% of T2DM and control patients, respectively. There was a strong association between the alleles (C vs. T: OR—3.32 [95% CI: 2.05–5.39]; *p* < 0.0001). The rs352140 SNP A and G alleles were present in A: 46.5%/G: 53.5% and A: 52.5%/G: 47.5% of T2DM and control patients, respectively. We did not find a significant association between the rs352140 SNP present in G and A alleles (A vs. G: OR—0.78 [95% CI: 0.53–1.16]; *p* = 0.23). The rs5743836 C and T alleles were found in C: 18.5%/T: 81.5% and C: 6.5%/T: 93.5% of the T2DM and control patients, respectively. There was a significant association between alleles (C vs. T: OR—3.26 [95% CI: 1.67–6.35]; *p* = 0.0002).

### 3.5. Multiple Linear Regression Analysis Studied between Three SNPs and T2DM Covariates

Table 6 represents the association between the 3 SNPs and 13 T2DM covariates such as age, weight, BMI, waist, hip, SBP, DBP, FBG, HbA1c, TC, TG, HDLC levels, and LDLC levels. The overall analysis of multiple linear models confirms that only DBP (*p* = 0.02) was associated (*p* > 0.05) with the SNPs in T2DM covariates.

### 3.6. ANOVA Analysis Studied between Three SNPs and T2DM Covariates

One-way ANOVA analysis was performed between the SNPs and the 13 covariates present in T2DM subjects (Table 7). In the rs187084 SNP, TG (2.29 ± 2.50) and HDLC (3.88 ± 0.74) were elevated in the TT genotypes. Age (63.68 ± 6.75), hip (106.70 ± 13.33), SBP (126.68 ± 11.22), DBP (80.28 ± 5.65), FBG (13.72 ± 3.34), HbA1c (7.33 ± 0.90), TC (6.12 ± 1.59), and HDLC (1.82 ± 1.68) levels were found to be high in TC genotypes. In the CC genotypes, weight (83.70 ± 12.84), BMI (31.93 ± 4.51), and waist (100.02 ± 9.40) were elevated. ANOVA analysis confirmed that TC (*p* = 0.03) and HDLC (*p* = 0.04) were significantly associated (*p* < 0.05). ANOVA analysis of the rs352140 SNP revealed increased weight (80.17 ± 17.44), waist (101.65 ± 22.58), hip (105.79 ± 13.77), FBG (13.72 ± 3.34), TC (2.67 ± 2.95), and TG (5.81 ± 1.01) in GG genotype. In the GA genotype, age (63.53 ± 7.65), BMI (31.73 ± 5.30), and SBP (127.70 ± 11.75) were high. In the AA genotype, DBP (81.77 ± 5.97), HbA1c (7.51 ± 0.86), and HDLC (1.57 ± 1.13) were elevated. ANOVA analysis of the rs352140 SNP showed an association between GG, GA, and AA genotypes and DBP (*p* = 0.0003), HbA1c (*p* = 0.03), and HDLC (*p* = 0.04). In the rs5743836 TT genotype, weight (80.74 ± 13.41), waist (99.38 ± 14.81), and FBG (13.41 ± 4.52) were elevated. In the TC genotype, BMI (31.59 ± 5.96), hip (104.67 ± 11.95), DBP (80.20 ± 6.29), HbA1c (7.21 ± 1.01), TG (2.39 ± 2.18), and HDLC (1.62 ± 1.74) were elevated. In the CC genotype, age (63.82 ± 7.52), SBP (123.50 ± 12.91), TC (5.95 ± 0.87), and LDLC (3.99 ± 0.51) were elevated. The ANOVA analysis confirmed that waist was significantly associated (*p* = 0.02) with the three genotypes.

### 3.7. Haplotype Analysis in SNPs Present in the TLR9 Gene

In the present study, seven haplotypes based on the three SNPs were constructed and analyzed to determine association with T2DM (Table 8). The C-G-T haplotype was found to be approximately four-fold higher in the T2DM group; the association was statistically significant (OR—4.95 [95% CI 1.16–103.4]; *p* < 0.003). These results indicate that the C-G-T haplotype confers risk towards T2DM. In addition, analysis of T2DM in combination with three SNPs in the *TLR9* gene showed that the C-G-C, C-A-T haplotypes were found to offer T2DM protection (OR—0.12.95 [95% CI 10.11–0.13]; *p* < 0.001 and OR—0.27 [95% CI 0.09–0.79]; *p* < 0.001).

### 3.8. Combination of LD Analysis with SNPs Present in the TLR9 Gene

Linkage disequilibrium analysis (LD), defined by the delta coefficient (D), of the three SNPs (i.e., rs352140, rs187084, and rs5743836 in the *TLR9* gene) was determined for both T2DM and control groups. No linkage disequilibrium was observed between the three SNPs, as shown in Table 9 and Figure 3. Due to the limited sample size, we could not identify the genetic marker that tags the actual causal variants in developing T2DM in the Saudi population.

### 3.9. MDR and GMDR Interaction Analysis in T2DM Cases

Gene–gene interaction was studied in 100 T2DM cases, and it was carried out in the SNPs (rs187084, rs352140 & rs5743836) in the *TLR9* gene (Table 10). The presentation of R1, R2, and R3 confirms a strong association between rs187084, rs352140, and rs5743836 (*p* < 0.001), with the combination of testing accuracy among S1 (T = 0.645), S1/S3 (T = 0.7), and S1/S2/S3 (T = 0.705). In addition, the MDR analysis (Figure 4) ensures the graphical representation model in women through S1–S3, as well as the findings from the study, which confirmed a graphical representation of the integrated effect of the complete loci models as high- and low-risk categories and statistical interactions. The dendrogram imposed the best feasible combination of synergistic/redundant SNP interaction. Among the three SNPs, S1 and S3 in the *TLR9* gene belonged to one cluster with substantial synergy and independent effect. A synergistic relationship was also identified between the S2 and S1 combinations. This supports the view that T2DM is inherited in a multifactorial manner, with major/minor gene(s) involvement, in addition to environmental factors.

## 4. Discussion

Toll-like receptors are found on cells of the innate immune system and contribute to the diagnosis of T2DM by identifying damage-associated molecular patterns. Upon TLR recognition, innate immune cells perform a number of actions, such as phagocytosis, cytokine synthesis, co-stimulatory molecule expression, and adhesion molecule expression [23]. TLRs are pivotal in the development of diabetes, which is primarily expressed in β-cells. TLR expression in β-cells provides an internal cell mechanism for innate signals regulating adaptive immune responses, and TLR signaling has been shown to be important in β-cell growth and activation [24]. The relationship between the *TLR9* gene and β-cells is considered to be the specialized cells in the pancreas, which is produced via insulin and also plays a major role in elevating blood glucose levels. However, this is not directly connected with immune cells. The connection between TLR9 and the innate immune system may be influenced by β-cell function and diabetes [14,25]. One of the major reasons for the increase in diabetes in Saudi Arabia is the burden of obesity/BMI and an aging population [5]. Consequently, cardiovascular diseases (CVD) have become the future health concern in Saudi Arabia [26].

We conducted a case-control study among Saudi Arabian T2DM patients and determined the prevalence of three SNPs (rs187084, rs352140, and rs5743836) in the study population. We confirmed a strong association between T2DM and rs187084 and rs5743836 SNPs. In addition, a negligible association was observed between T2DM and the rs352140 SNP. Additionally, DBP levels (*p* = 0.02), TC (*p* = 0.03), and LDLC (*p* = 0.04) were significantly associated with rs187084. DBP (*p* = 0.0003), Hb1Ac (*p* = 0.03), and HDLC (*p* = 0.04) were significantly associated with rs352140, and waist (*p* = 0.02) was significantly associated with rs5743836. The C-G-T haplotype increased the risk for developing T2DM (*p* < 0.003) by approximately four-fold, and the combination of C-G-C/C-A-T haplotypes showed protective role towards T2DM. LD analysis showed a negative association, and both MDR/GMDR analyses showed strong association of gene–gene interaction, dendrogram, and graphical representation (*p* < 0.05) with T2DM cases. Overall, we confirmed that T2DM development is associated with SNPs present in the *TLR9* gene in the Saudi population.

TLRs are widely expressed in the human body cells, and 12 SNPs are present in the *TLR9* gene. rs187084 is present at the promoter region [27]. In this study, 23% of CC genotypes, 25% of AA genotypes, and 11% of CC genotypes were present in three different SNPs in the form of homozygous variants in T2DM patients, whereas, in controls, 8%, 23%, 4% were present. This indicates homozygous variants are high in the Saudi Arabian population. However, rs352140 prevalence was similar between both groups. Only the co-dominant model showed a statistical association (*p* = 0.02), with 1.9 times increased risk.

A previous study in an Egyptian population has confirmed a significant role of rs5743836 in T2DM (*p* = 0.01) and diabetes foot (DF) (*p* = 0.02). In T2DM patients, 73.3% and 6.7% of the heterozygous genotypes were observed. In DF patients, TT and TC genotypes were present (50% each), and the CC genotype was not observed. The genotype frequencies observed in our study were different and could be due to a number of factors. First, the Egyptian study contained a small sample size. In addition, differences could be due to the ethnic diversity between the populations [12]. Another study was carried out in Chinese patients diagnosed with T2DM, CAD, and a combination of T2DM + CAD. The study determined negative associations in T2DM, CAD, and a combination of T2DM + CAD among −1486T>C and −1237T>C SNPs. However, this study was documented over a decade ago [17]. The rs187084 SNP was studied in other human diseases with different ethnicities, and no significant associations were determined between the SNP and disease development [28,29]. However, a positive association was confirmed at the +1174 locus [29]. In the Iranian population, the rs352140 SNP was not associated with T2DM or DN [13]; however, this could be a result of small sample size. The rs352140 SNP was studied in Chinese children with T1DM, and a strong association was observed between the SNP and T1DM development [11].

Different meta-analysis studies were reported with rs187084, rs352140, and rs5743836 SNPs in different human diseases, but there are no meta-analysis studies documented between rs187084, rs352140, and rs5743836 SNPs in the *TLR9* gene and T2DM.

Family history is considered as one of the non-modifiable risk factors along with age, gender, and height. Family of T2DM plays a significant role in the Saudi Arabian population. The first degree relative has 30–70% risk of developing T2DM disease. Family history of T2DM has been shown to increase disease risk by two-fold in the future generations [30]. Tuomilehto et al. [31] confirmed effectiveness of an intervention program, following that the intervention program lowers T2DM from 58% to 43% in a span of three years. In Saudi Arabia, the combination of family history of T2DM and parental consanguinity will lead to increase the risk of both impaired fasting glucose and diabetes in the offspring [32]. Nonetheless, frequently checking glucose levels, altering eating habits, and increasing physical exercise can alter the quality of insulin produced in order to fulfill the body’s blood glucose targets and maintain appropriate blood glucose levels.

HTN (both essential and idiopathic) is defined as a chronic condition marked by expanded SBP and DBP levels. In our study, we have found an association between DBP and the *TLR9* gene. The DBP was found to be higher in the T2DM patients (*p* = 0.002). These results were confirmed with multiple linear regression; a positive association was observed between DBP and T2DM (*p* = 0.02). The association between the rs352140 SNP and DBP was statistically significant (*p* = 0.0003). The elevated levels of DBP were found in the rs352140 AA genotype with 81.77 ± 5.97, which indicates there is a relation between HTN and the *TLR9* gene. The relationship between the *TLR9* gene and HTN has been previously documented [33,34]. Overall, based on the current study results, we conclude that the *TLR9* gene plays a role in T2DM and HTN development in the Saudi Arabian population.

The major limitation of this study was small population size (100 T2DM patients and 100 healthy controls). The other limitations of this study could be not matching with gender and anthropometric characteristics in T2DM cases and control subjects. However, the number of participants included in this study was based on a sample size calculation. Only three SNPs were screened in this study. Another limitation of this study was not selecting T2DM patients with any of the infectious diseases. The final limitation of this study was missing of functional studies in TLR9. The strength of this study was enrolling T2DM patients with a family history of diabetes. By selecting the patients with complete family history, we were able to determine the molecular role of the *TLR9* gene. Additional statistics was another strength.

## 5. Conclusions

In this study, we aimed to understand the molecular role of rs187084, rs352140, and rs5743836 SNPs in the *TLR9* gene in T2DM. Previously, different SNPs in the *TLR9* gene were well-established with infectious diseases and cancers. Presently, a link has been established between infectious diseases and T2DM [35] as well as T2DM and cancers [36]. We confirm that rs187084 and rs5743836 are strongly associated with T2DM and rs352140 is nominally associated with T2DM. Further case-control studies, which include various ethnicities and factor gene-environment interactions, are required to confirm our study results. However, this study serves as a foundational study for the use of rs187084 and rs5743836 as molecular targets for T2DM diagnosis and therapies. In addition, we found that HTN is associated with T2DM. Further molecular studies in the Saudi population among HTN patients is required to confirm the role SNPs present in the *TLR9* gene in HTN development.

## Figures and Tables

**Figure 1 biology-12-01439-f001:**
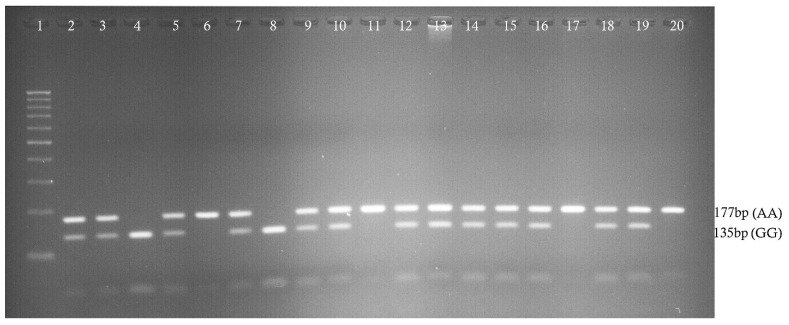
Agarose gel electrophoresis confirmed rs352140 SNP. Lane 1 indicates 100 bp ladder. Lanes 4 and 8 indicate GG (135/42 bp) genotypes. Lanes 2, 3, 5, 7, 8, 9, 12–16, and 18–19 indicate GA (177/135/42 bp) genotypes. Lanes 6, 11, 17, and 20 indicate AA (177 bp) genotypes.

**Figure 2 biology-12-01439-f002:**
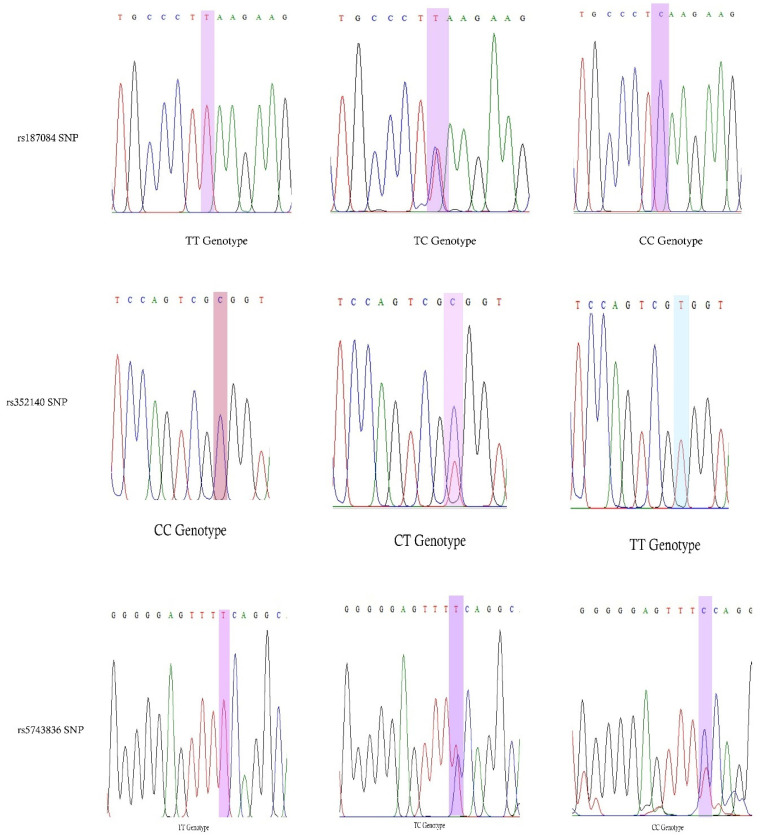
Sanger sequencing analysis revealed probable genotypes for rs187084 and rs5743836 SNPs, and PCR–RFLP analysis revealed rs352140 SNP genotypes.

**Figure 3 biology-12-01439-f003:**
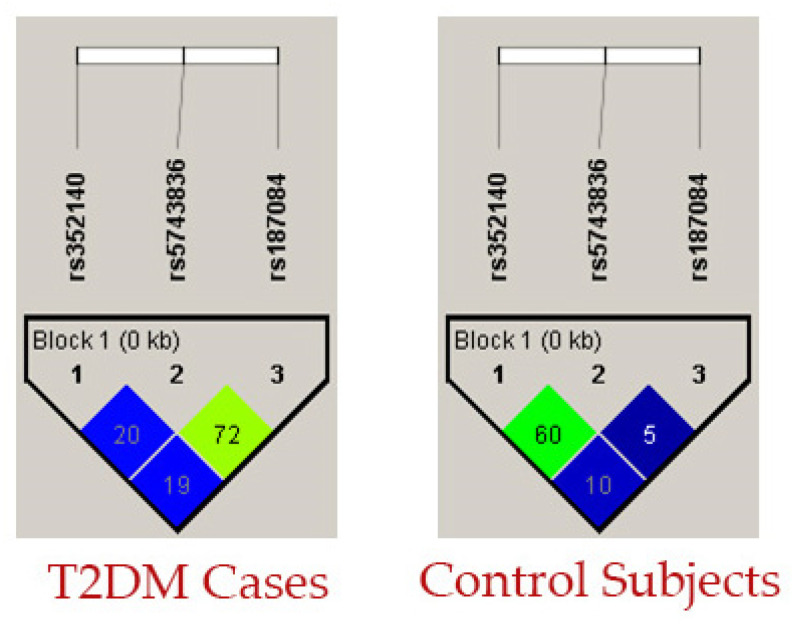
Analysis of LD in T2DM cases and controls using three SNPs in the *TLR9* gene.

**Figure 4 biology-12-01439-f004:**
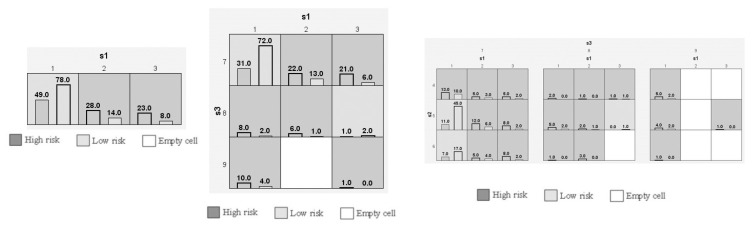
Analysis of T2DM risk factors by multifactor dimensionality reduction, represented graphically with dark and light cells.

**Figure 5 biology-12-01439-f005:**
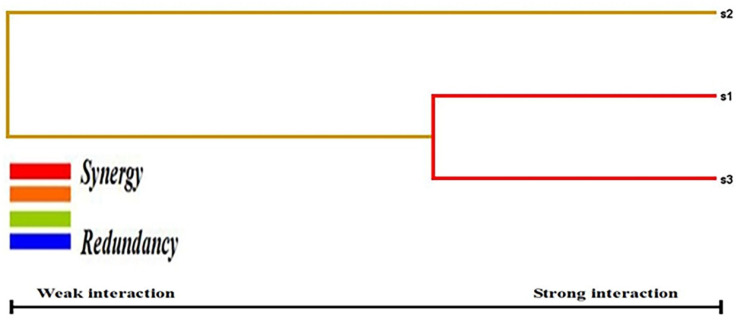
Dendrogram analysis in T2DM patients using MDR.

**Table 1 biology-12-01439-t001:** Details of SNPs involved in this study.

Gene	Rs Number	Position	Forward Primer	Reverse Primer	PCR	Temp	Enzyme	RFLP Band Sizes
*TLR9*	rs187084	−1486T>C	TTCATTCATTCAGCCTTCACTCA	GAGTCAAAGCCACAGTCCACA	565 bp	68 °C	Hpy188III	T-288/277; C-288/150/127 bp
*TLR9*	rs352140	G2848A	AAGCTGGACCTCTACCACGA	TTGGCTGTGGATGTTGTT	177 bp	60 °C	BstUI	G-135/42 bp; A-177 bp
*TLR9*	rs5743836	−1237T>C	GCTGGATGGCCCTGTTGA	GCCTCAGGGCCTTGGGAT	120 bp	68 °C	-	-

**Table 2 biology-12-01439-t002:** Characteristics of study participants.

	Controls (n = 100)	T2DM (n = 100)	*p*-Value
Age (Years)	52.12 ± 7.31	63.27 ± 8.37	<0.0001
Gender (Female: Male)	54%:46%	37%:63%	0.03
Weight (Kilograms)	76.39 ± 12.21	79.88 ± 13.79	0.05
Height (Centimeters)	161.39 ± 7.23	159.60 ± 8.58	0.85
BMI (kg/m^2^)	29.34 ± 4.42	31.30 ± 5.18	0.004
Waist (cm)	97.64 ± 18.68	96.58 ± 17.12	0.67
Hip (cm)	103.66 ± 15.70	104.18 ± 16.11	0.81
SBP (mmHg)	116.51 ± 6.34	126.30 ± 10.75	<0.0001
DBP (mmHg)	77.13 ± 5.43	79.76 ± 6.42	0.002
FBG (mmol/L)	5.44 ± 0.47	12.99 ± 4.38	<0.0001
HbA1c	5.35 ± 0.42	7.17 ± 0.87	<0.0001
TC (mmol/L)	5.17 ± 0.93	5.76 ± 1.20	0.001
TG (mmol/L)	1.82 ± 1.01	2.11 ± 1.99	0.19
HDLC (mmol/L)	0.67 ± 0.29	1.37 ± 1.20	<0.0001
LDLC (mmol/L)	3.68 ± 0.80	3.80 ± 0.98	0.34
Family History of Diabetes	43 (43%)	100 (100%)	<0.0001

**Table 3 biology-12-01439-t003:** Analysis of Hardy Weinberg equilibrium studies in the three SNPs in the *TLR9* gene.

TLR9 (SNPs)	T2DM Cases (n = 100)			Control Subjects (n = 100)		
	Genotypes	ꭓ^2^	*p*-Value	Genotypes	ꭓ^2^	*p*-Value
rs187084	74%/15%/11%	6.08	*p* = 0.02	91%/5%/4%	34.6	*p* < 0.01
rs352140	32%/43%/25%	1.29	*p* = 0.25	18%/59%/23%	10.7	*p* < 0.001
rs5743836	49%/28%/23%	15.9	*p* = 0.02	78%/14%/8%	38	*p* < 0.01

**Table 4 biology-12-01439-t004:** Genotype frequencies in rs187084, rs352140, and rs5743836 SNPs in T2DM and control subjects.

Gene (rs Number)	Genotypes	Controls (n = 100)	T2DM (n = 100)	OR (95% CI) and *p*-Value
*TLR9* (rs187084)	TT	78 (78%)	49 (49%)	
	TC	14 (14%)	28 (28%)	OR-3.18 [95% CI: 1.52–6.63]; *p* = 0.001
	CC	08 (08%)	23 (23%)	OR-4.57 [95% CI: 1.89–11.04]; *p* = 0.0003
	TC + CC vs. TT	22 (22%)	51 (51%)	OR-3.69 [95% CI: 1.99–6.82]; *p* = 0.0002
	TT + CC vs. TC	86 (86%)	72 (72%)	OR-0.41 [95% CI: 0.20–0.85]; *p* = 0.01
	TT + TC vs. CC	92 (92%)	77 (77%)	OR-0.29 [95% CI: 0.12–0.68]; *p* = 0.003
*TLR9* (rs352140)	GG	18 (18%)	32 (32%)	
	GA	59 (59%)	43 (43%)	OR-0.41 [95% CI: 0.21–1.82]; *p* = 0.01
	AA	23 (23%)	25 (25%)	OR-0.61 [95% CI: 0.27–1.37]; *p* = 0.23
	GA + AA vs. GG	82 (82%)	68 (68%)	OR-0.46 [95% CI: 0.24–0.91]; *p* = 0.02
	GG + AA vs. GA	41 (41%)	57 (57%)	OR-1.91 [95% CI: 1.08–3.34]; *p* = 0.02
	GG + GA vs. AA	77 (77%)	75 (75%)	OR-0.89 [95% CI: 0.46–1.71]; *p* = 0.74
*TLR9* (rs5743836)	TT	91 (91%)	74 (74%)	
	TC	05 (5%)	15 (15%)	OR-3.68 [95% CI: 1.28–10.62]; *p* = 0.01
	CC	04 (4%)	11 (11%)	OR-3.38 [95% CI: 1.03–11.07]; *p* = 0.03
	TC + CC vs. TT	09 (09%)	26 (26%)	OR-3.55 [95% CI: 1.56–8.04]; *p* = 0.001
	TT + CC vs. TC	95 (95%)	85 (85%)	OR-0.29 [95% CI: 0.10–0.85]; *p* = 0.01
	TT + TC vs. CC	96 (96%)	89 (89%)	OR-0.33 [95% CI: 0.10–1.09]; *p* = 0.06

**Table 5 biology-12-01439-t005:** Allele frequencies in rs187084, rs352140, and rs5743836 SNPs in T2DM and control subjects.

Gene (rs Number)	Alleles	Controls (n = 100)	T2DM (n = 100)	OR (95% CI) and *p*-Value
*TLR9* (rs187084)	T	170 (85%)	126 (63%)	
	C	30 (15%)	74 (37%)	OR-3.32 [95% CI: 2.05–5.39]; *p* < 0.0001
*TLR9* (rs352140)	G	95 (47.5%)	107 (53.5%)	
	A	105 (52.5%)	93 (46.5%)	OR-0.78 [95% CI: 0.53–1.16]; *p* = 0.23
*TLR9* (rs5743836)	T	187 (93.5%)	163 (81.5%)	
	C	13 (6.5%)	37 (18.5%)	OR-3.26 [95% CI: 1.67–6.35]; *p* = 0.0002

**Table 6 biology-12-01439-t006:** Multiple linear regression analysis of rs187084, rs352140, and rs5743836 SNPs and T2DM covariates.

Covariates	R-Value	Adjusted R Square Value	Standardized β-Coefficient for rs187084	Standardized β-Coefficient for rs352140	Standardized β-Coefficient for rs57438386	F	*p*-Value
Age	0.054	−0.028	−0.014	−0.034	−0.033	0.094	0.963
Weight	0.120	−0.017	0.044	−0.044	−0.097	0.464	0.708
BMI	0.119	−0.017	−0.010	−0.117	−0.012	0.461	0.710
Waist	0.176	0.001	0.032	−0.069	−0.157	1.01	0.388
Hip	0.104	−0.020	0.103	−0.027	0.060	0.351	0.788
SBP	0.145	−0.010	0.016	−0.042	−0.137	0.687	0.562
DBP	0.306	0.066	−0.037	0.310	−0.012	3.314	0.023
FBG	0.166	−0.003	0.003	−0.042	−0.163	0.910	0.493
Hb1Ac	0.164	−0.003	0.111	0.105	0.015	0.886	0.451
TC	0.188	0.005	−0.165	−0.034	−0.127	1.168	0.326
TG	0.180	0.002	−0.119	−0.117	−0.005	1.074	0.364
HDLC	0.221	0.019	0.082	0.193	0.020	1.646	0.184
LDLC	0.176	0.001	−0.136	−0.047	−0.136	1.025	0.385

**Table 7 biology-12-01439-t007:** ANOVA analysis between SNPs and anthropometric/biochemical parameters in T2DM patients.

	rs187084				rs352140				rs5743836			
	TT (n = 49)	TC (n = 28)	CC (n = 23)	*p*-Value	GG (n = 32)	GA (n = 43)	AA (n = 25)	*p*-Value	TT (n = 74)	TC (n = 15)	CC (n = 11)	*p*-Value
Age	63.35 ± 9.23	63.68 ± 6.75	62.61 ± 8.52	0.90	63.47 ± 8.95	63.53 ± 7.65	62.56 ± 9.06	0.88	63.08 ± 8.64	63.80 ± 8.06	63.82 ± 7.52	0.93
Weight	80.28 ± 15.52	76.04 ± 10.32	83.70 ± 12.84	0.13	80.17 ± 17.44	80.13 ± 10.54	79.06 ± 13.98	0.94	80.74 ± 13.41	77.78 ± 14.37	76.95 ± 16.04	0.57
BMI	31.83 ± 5.74	29.83 ± 4.50	31.93 ± 4.51	0.21	31.72 ± 5.55	31.73 ± 5.30	30.02 ± 4.42	0.36	31.30 ± 5.04	31.59 ± 5.96	30.87 ± 5.56	0.94
Waist	98.33 ± 18.36	91.55 ± 18.88	100.02 ± 9.40	0.14	101.65 ± 22.58	96.10 ± 9.62	92.13 ± 21.23	0.12	99.38 ± 14.81	87.17 ± 22.97	92.80 ± 16.35	0.02
Hip	102.55 ± 19.77	106.70 ± 13.33	104.07 ± 12.74	0.57	105.79 ± 13.77	103.45 ± 18.44	103.88 ± 14.28	0.81	104.37 ± 17.31	104.67 ± 11.95	101.20 ± 15.35	0.82
SBP	126.02 ± 10.19	126.68 ± 11.22	126.40 ± 11.80	0.96	126.36 ± 9.92	127.70 ± 11.75	123.68 ± 9.68	0.33	127.55 ± 10.72	122.07 ± 8.78	123.50 ± 12.91	0.13
DBP	79.68 ± 7.19	80.28 ± 5.65	79.25 ± 5.91	0.89	76.00 ± 6.34	80.45 ± 5.64	81.77 ± 5.97	0.0003	79.94 ± 6.72	80.20 ± 6.29	77.78 ± 4.41	0.56
FBG	12.65 ± 4.68	13.72 ± 3.34	12.84 ± 4.87	0.58	13.11 ± 3.60	13.05 ± 5.32	12.74 ± 3.55	0.94	13.41 ± 4.52	11.96 ± 3.41	11.61 ± 4.36	0.27
Hb1Ac	7.03 ± 0.86	7.33 ± 0.90	7.26 ± 0.85	0.29	7.18 ± 0.72	6.95 ± 0.94	7.51 ± 0.86	0.03	7.18 ± 0.89	7.21 ± 1.01	7.04 ± 0.61	0.87
TC	5.79 ± 1.00	6.12 ± 1.59	5.24 ± 0.90	0.03	2.67 ± 2.95	1.75 ± 0.95	2.00 ± 1.68	0.13	5.82 ± 1.26	5.33 ± 1.07	5.95 ± 0.87	0.30
TG	2.29 ± 2.50	2.24 ± 1.60	1.54 ± 0.74	0.31	5.81 ± 1.01	5.79 ± 1.23	5.63 ± 1.40	0.82	2.07 ± 2.06	2.39 ± 2.18	1.98 ± 1.22	0.83
HDLC	1.12 ± 0.87	1.82 ± 1.68	1.32 ± 0.95	0.04	0.95 ± 0.59	1.56 ± 1.48	1.57 ± 1.13	0.04	1.38 ± 1.10	1.62 ± 1.74	0.95 ± 0.92	0.36
LDLC	3.88 ± 0.74	3.84 ± 1.44	3.60 ± 0.76	0.52	3.84 ± 0.93	3.86 ± 1.01	3.66 ± 1.02	0.70	3.86 ± 0.97	3.39 ± 1.21	3.99 ± 0.51	0.19

**Table 8 biology-12-01439-t008:** Haplotype association between TLR9 variants and T2DM patients. * indicates the significant association.

S. No	rs187084	rs352140	rs5743836	Frequency	OR (95% CI)	*p*-Value
1	T	A	T	0.3215	1.00	---
2	T	G	T	0.3114	0.86 (0.39–1.89)	0.72
3	C	A	T	0.1324	0.27 (0.09–0.79)	<0.001 *
4	C	G	T	0.1098	4.95 (1.16–103.64)	<0.003 *
5	T	G	C	0.0725	2.41 (0.82–7.09)	0.11
6	T	A	C	0.0347	1.39 (0.54–3.60)	0.5
7	C	G	C	0.0114	0.12 (0.11–0.13)	<0.001 *

**Table 9 biology-12-01439-t009:** Linkage disequilibrium analysis in T2DM and healthy controls with the three SNPs in the *TLR9* gene.

Participants	L1	L2	D’	r^2^	Participants	L1	L2	D’	r^2^
T2DM Cases	rs352140	rs5743836	0.205	0.008	Controls	rs352140	rs5743836	0.602	0.028
T2DM Cases	rs352140	rs187084	0.199	0.027	Controls	rs352140	rs187084	0.109	0.002
T2DM Cases	rs5743836	rs187084	0.723	0.07	Controls	rs5743836	rs187084	0.058	0.001

**Table 10 biology-12-01439-t010:** Gene–gene interaction in determining T2DM risk.

Model No.	Genes Included in Best Combination in Each Model	Training Accuracy	Testing Accuracy	CVC	*p*-Value
1	S1 (rs187084)	0.645	0.645	10/10	<0.001
2	S1, S3 (rs187084, rs5743836)	0.71	0.7	10/10	<0.001
3	S1, S2, S3 (rs187084, rs352140, rs5743836)	0.7261	0.705	10/10	<0.001

## Data Availability

All the data has been presented in the manuscript.

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
