# Peer review of "Toll-like Receptor 9 Gene in the Development of Type 2 Diabetes Mellitus in the Saudi Arabian Population"

_biology, 2023, doi:10.3390/biology12111439_

Round 1
Reviewer 1 Report
Comments and Suggestions for Authors
Major comments:
1) Avoid the use of terms as “diabetic or prediabetic individuals”. Use instead: “people with diabetes / patients / individuals with diabetes” (Diabetes Res Clin Pract 2012; 97: 425-431).
2) In the background section, I suggest explaining why the authors selected those specific SNPs of TLR9 gene. Is there previous evidence of its relevance?
3) I suggest that the model presented is not case-control. It seems two different cohort. A well performed case-control study should be done considering: sex, BMI and age (at least). If not, justify the relevance of your model or its impact into your results.
4) In my opinion is a big mistake only considering fasting glucose levels to considere those patients as “without diabetes”. Furthemore, current guidelines recomend a second probe to confirm T2D diagnosis.
5) Please, considere summarizing your results since most of them are reported as table and text form.
6) Also summarize your discussion. Avoid repeating ideas from background section.
Minor comments:
1) In the abstract section: suprress the result regarding T2DM and anthropometric and biochemical associations. This has already been fully studied and it is not a main result of your manuscrupt.
2) The references 18 and 19 are not justified. Both are not part of the main objective of the study.
Comments on the Quality of English LanguageThe quality of English is fine. No improvements are required in this area.
Author Response
Major comments:
1) Avoid the use of terms as “diabetic or prediabetic individuals”. Use instead: “people with diabetes / patients / individuals with diabetes” ().
A) We have updated the diabetic individuals with individuals with diabetes in the revised manuscript and we have added the recommended citation.
2) In the background section, I suggest explaining why the authors selected those specific SNPs of TLR9 gene. Is there previous evidence of its relevance?
A) The 3 SNPs (rs187084, rs352140 and rs5743836) were selected based on the documentation of the previous studies in different types of diabetes such as T1DM [1], T2DM [2], T2DM+CAD [3] and diabetic foot [2]. However, previous studies were documented in Saudi Arabia with rs187084 [4], and rs352140 [5] SNPs in different human diseases [4,5] excluding T2DM. However, rs5743836 SNP was not documented in the Saudi population, but it was studied in T2DM patient [2]. So, studying rs187084, rs352140 and rs5743836 SNPs in Saudi patients with T2DM will be an interesting and initial study to document the role of rs187084, rs352140 and rs5743836 SNPs in T2DM. Since a decade and above, the prevalence of T2DM was expanding in Saudi Arabia. Based on these factors, this study was designed and we have added references and discussed in the introduction of the revised manuscript with the 3 SNPs of the TLR9 gene and with different types of diabetes and probable role.
3) I suggest that the model presented is not case-control. It seems two different cohort. A well performed case-control study should be done considering: sex, BMI and age (at least). If not, justify the relevance of your model or its impact into your results.
A) We have designed this study as a case-control study based on the sample size calculation. The details of the sample size calculation were added in the revised manuscript under the sub-section of 2.2 in the materials and methods.
4) In my opinion is a big mistake only considering fasting glucose levels to consider those patients as “without diabetes”. Furthermore, current guidelines recommend a second probe to confirm T2D diagnosis.
A) All T2DM cases and controls were selected from Diabetic clinic of university hospital. All the individuals were on regular follow-up. All T2DM patients have crossed a minimum of 5 years of developing T2DM (as per medical records). Based on WHO criteria, we have documented to confirm the T2DM patients using FBG levels.
5) Please, consider summarizing your results since most of them are reported as table and text form.
A) The results were already categorized as 3.1 to 3.9 in the main document of the results section. However, we have described in second paragraph of the Discussion. If you have any suggestion to describe them, kindly let us know.
6) Also summarize your discussion. Avoid repeating ideas from background section.
A) Discussion was revised and we have done our best in shortening it.
Minor comments:
1) In the abstract section: suppress the result regarding T2DM and anthropometric and biochemical associations. This has already been fully studied and it is not a main result of your manuscript.
A) We have updated in the revised manuscript.
2) The references 18 and 19 are not justified. Both are not part of the main objective of the study.
A) We have deleted references 18 and 19 in the revised manuscript.
References
- Wang, Y.; Xia, Y.; Chen, Y.; Xu, L.; Sun, X.; Li, J.; Huang, G.; Li, X.; Xie, Z.; Zhou, Z. Association analysis between the TLR9 gene polymorphism rs352140 and type 1 diabetes. Frontiers in Endocrinology 2023, 14.
- Wifi, M.-N.A.; Assem, M.; Elsherif, R.H.; El-Azab, H.A.-F.; Saif, A. Toll-like receptors-2 and-9 (TLR2 and TLR9) gene polymorphism in patients with type 2 diabetes and diabetic foot. Medicine 2017, 96.
- Liu, F.; Lu, W.; Qian, Q.; Qi, W.; Hu, J.; Feng, B. Frequency of TLR 2, 4, and 9 gene polymorphisms in Chinese population and their susceptibility to type 2 diabetes and coronary artery disease. Journal of Biomedicine and Biotechnology 2012, 2012.
- Semlali, A.; Parine, N.R.; Al Amri, A.; Azzi, A.; Arafah, M.; Kohailan, M.; Shaik, J.P.; Almadi, M.A.; Aljebreen, A.M.; Alharbi, O. Association between TLR-9 polymorphisms and colon cancer susceptibility in Saudi Arabian female patients. OncoTargets and therapy 2016, 1-11.
- Eed, E.M.; Hawash, Y.A.; Khalifa, A.S.; Alsharif, K.F.; Alghamdi, S.A.; Almalki, A.A.; Almehmadi, M.M.; Ismail, K.A.; Taha, A.A.; Saber, T. Association of toll-like receptors 2, 4, 9 and 10 genes polymorphisms and Helicobacter pylori-related gastric diseases in Saudi patients. Indian journal of medical microbiology 2020, 38, 94-100.
Reviewer 2 Report
Comments and Suggestions for Authors
The manuscript in question explores the association between TLR9 polymorphisms and T2D in Saudi Arabia population. While TLR9 polymorphisms and their associations to diabetes has been described before, this is the first report from Saudi Arabia population and adds more evidence to the literature regarding significance of studying SNPs in TLR9. I only have minor comments to the authors which may help to improve the overall mauscript-
1. Did the authors determine the functional consequence of these SNPs? Do these SNPs in TLR9 affect its ability to respond to TLR9 agonists? downstream activation of inflammatory mediators like NF-kB?
2. Can the authors comment on whether TLR9 is expressed in beta cells?
3. The SNPs described in the paper have been previously associated with a lot of other disease states like SLE, malaria etc, can the authors comment on why this SNP is associated with such different disease susceptibilities.
Comments on the Quality of English Language
Minor english editing required
Author Response
The manuscript in question explores the association between TLR9 polymorphisms and T2D in Saudi Arabia population. While TLR9 polymorphisms and their associations to diabetes has been described before, this is the first report from Saudi Arabia population and adds more evidence to the literature regarding significance of studying SNPs in TLR9. I only have minor comments to the authors which may help to improve the overall manuscript-
A) Dear Reviewer, Thank you for your valuable comment. We have justified all the raised query.
- Did the authors determine the functional consequence of these SNPs? Do these SNPs in TLR9 affect its ability to respond to TLR9 agonists? downstream activation of inflammatory mediators like NF-kB?
- A) This is one of the interesting questions. In reality, this study was designed to screen the single nucleotide polymorphisms (SNPs) in the TLR9 gene among T2DM patients. Now, we have an intention to perform the functional studies in the future to rule out the role of TLR9 respondent towards agonists downstream activation of inflammatory mediators like NF-kB. We have described this as one of the limitations of this study.
- Can the authors comment on whether TLR9 is expressed in beta cells?
- A) We have explained in the discussion of the revised manuscript about TLR9 in the β-cells.
- The SNPs described in the paper have been previously associated with a lot of other disease states like SLE, malaria etc, can the authors comment on why this SNP is associated with such different disease susceptibilities.
- A) The TLR9 gene is mainly associated with chronic autoimmune diseases in which systemic lupus erythematosus (SLE) is one of them. Both TLR9 and SLE is connected via immune system and TLR9 contributes majorly towards the progression and development of SLE. The connection between TLR9 and SLE is connected with immune cells of endosomes in which dendritic and β-cells are present. Additionally, TLR9 was also studied in atopic dermatitis [1], pulmonary tuberculosis [2], cancers [3-5], SLE [6], Asthma [7], malaria [8,9] and different forms of diabetes [10-13]. The TLR9 gene has a role with the above-mentioned diseases are ultimately SNPs will also have a role towards the human diseases associated with TLR9
References
- Zhou, B.; Liang, S.; Shang, S.; Li, L. Association of TLR2 and TLR9 gene polymorphisms with atopic dermatitis: a systematic review and meta-analysis with trial sequential analysis. Immunological Medicine 2023, 46, 32-44.
- Chen, Z.; Wang, W.; Liang, J.; Wang, J.; Feng, S.; Zhang, G. Association between toll-like receptors 9 (TLR9) gene polymorphism and risk of pulmonary tuberculosis: meta-analysis. BMC Pulmonary Medicine 2015, 15, 1-10.
- Zhang, L.; Qin, H.; Guan, X.; Zhang, K.; Liu, Z. The TLR9 gene polymorphisms and the risk of cancer: evidence from a meta-analysis. PloS one 2013, 8, e71785.
- Yang, S.; Liu, L.; Xu, D.; Li, X. The relationship of the TLR9 and TLR2 genetic polymorphisms with cervical cancer risk: a meta-analysis of case-control studies. Pathology & Oncology Research 2020, 26, 307-315.
- Wan, G.-X.; Cao, Y.-W.; Li, W.-Q.; Li, Y.-C.; Zhang, W.-J.; Li, F. Associations between TLR9 polymorphisms and cancer risk: evidence from an updated meta-analysis of 25,685 subjects. Asian Pacific Journal of Cancer Prevention 2014, 15, 8279-8285.
- Wang, D.; Zhang, C.; Zhou, Z.; Pei, F. TLR9 polymorphisms and systemic lupus erythematosus risk: an update meta-analysis study. Rheumatology International 2016, 36, 585-595.
- Lachheb, J.; Dhifallah, I.; Chelbi, H.; Hamzaoui, K.; Hamzaoui, A. Toll‐like receptors and CD14 genes polymorphisms and susceptibility to asthma in Tunisian children. Tissue Antigens 2008, 71, 417-425.
- Campino, S.; Forton, J.; Auburn, S.; Fry, A.; Diakite, M.; Richardson, A.; Hull, J.; Jallow, M.; Sisay-Joof, F.; Pinder, M. TLR9 polymorphisms in African populations: no association with severe malaria, but evidence of cis-variants acting on gene expression. Malaria Journal 2009, 8, 1-8.
- Dhangadamajhi, G.; Kar, A.; Rout, R.; Dhangadamajhi, P. A meta-analysis of TLR4 and TLR9 SNPs implicated in severe malaria. Revista da Sociedade Brasileira de Medicina Tropical 2017, 50, 153-160.
- Wang, Y.; Xia, Y.; Chen, Y.; Xu, L.; Sun, X.; Li, J.; Huang, G.; Li, X.; Xie, Z.; Zhou, Z. Association analysis between the TLR9 gene polymorphism rs352140 and type 1 diabetes. Frontiers in Endocrinology 2023, 14, 1030736.
- Zhao, J.; Zhang, L.-X.; Wang, Y.-T.; Li, Y.; Chen, M., Hong-Lin. Genetic polymorphisms and the risk of diabetic foot: a systematic review and meta-analyses. The International Journal of Lower Extremity Wounds 2022, 21, 574-587.
- Liu, F.; Lu, W.; Qian, Q.; Qi, W.; Hu, J.; Feng, B. Frequency of TLR 2, 4, and 9 gene polymorphisms in Chinese population and their susceptibility to type 2 diabetes and coronary artery disease. BioMed Research International 2012, 2012.
- Wifi, M.-N.A.; Assem, M.; Elsherif, R.H.; El-Azab, H.A.-F.; Saif, A. Toll-like receptors-2 and-9 (TLR2 and TLR9) gene polymorphism in patients with type 2 diabetes and diabetic foot. Medicine 2017, 96.
Round 2
Reviewer 1 Report
Comments and Suggestions for Authors
The authors have addressed almost all of my comments. However, in my opinion the study has not a case-control design, since it was not paired by sex or anthropometric characteristics. I did not understand their answer regarding this point.
Also, please note that the reference regarding correct language in diabetes, was only provided for author's knowledge. It could be not included into references list.
Comments on the Quality of English LanguageThere are some typos that requires correction.
Author Response
Dear Reviewer,
Once again, thank you for your valuable comments. We’re happy to receive the comments. Now, we have justified below and updated in the revised manuscript and highlighted with yellow color. If you feel, we haven’t justified properly or you have some other suggestions, kindly let us know prior to the acceptance. All the authors have discussed and finalized the comments and approved it. Once again, we thank you for your valuable comments towards our manuscript.
- Q) The authors have addressed almost all of my comments. However, in my opinion the study has not a case-control design, since it was not paired by sex or anthropometric characteristics. I did not understand their answer regarding this point.
- A) We have designed this as a case-control study based on inclusion and exclusion criteria as well as with the sample size calculation using the formula. However, we agree with your comment as none of the sex (gender) or anthropometric characteristics were matching and we have confirmed this as one of the limitations of this study.
- Q) Also, please note that the reference regarding correct language in diabetes, was only provided for author's knowledge. It could be not included into references list.
- A) We all the authors thankful for your valuable comment and now we have now removed the reference in the revised manuscript.